biomedical engineering/biophysics

diabetes mellitus, diabetic retinopathy, thrombosis, microaneurysm, computational fluid dynamics, force coupling method

**Authors for correspondence:**
George E. Karniadakis
e-mail: george_karniadakis@brown.edu
Jennifer K. Sun
e-mail: Jennifer.Sun@joslin.harvard.edu

†These authors contributed equally to this work.

# Predictive modelling of thrombus formation in diabetic retinal microaneurysms

He Li[1,†], Konstantina Sampani[2,3,†], Xiaoning Zheng[1,†], Dimitrios P. Papageorgiou[5], Alireza Yazdani[1], Miguel O. Bernabeu[6], George E. Karniadakis[1] and Jennifer K. Sun[2,4]

[1]Division of Applied Mathematics, Brown University, Providence, RI 02912, USA
[2]Beetham Eye Institute, Joslin Diabetes Center, Boston, MA, USA
[3]Department of Medicine, and [4]Department of Ophthalmology, Harvard Medical School, Boston, MA, USA
[5]Department of Materials Science and Engineering, Massachusetts Institute of Technology, Cambridge, MA, USA
[6]Centre for Medical Informatics, Usher Institute, University of Edinburgh, Edinburgh, UK

HL, 0000-0002-1735-8899

Microaneurysms (MAs) are one of the earliest clinically visible signs of diabetic retinopathy (DR). Vision can be reduced at any stage of DR by MAs, which may enlarge, rupture and leak fluid into the neural retina. Recent advances in ophthalmic imaging techniques enable reconstruction of the geometries of MAs and quantification of the corresponding haemodynamic metrics, such as shear rate and wall shear stress, but there is lack of computational models that can predict thrombus formation in individual MAs. In this study, we couple a particle model to a continuum model to simulate the platelet aggregation in MAs with different shapes. Our simulation results show that under a physiologically relevant blood flow rate, thrombosis is more pronounced in saccular-shaped MAs than fusiform-shaped MAs, in agreement with recent clinical findings. Our model predictions of the size and shape of the thrombi in MAs are consistent with experimental observations, suggesting that our model is capable of predicting the formation of thrombus for newly detected MAs. This is the first quantitative study of thrombosis in MAs through simulating platelet aggregation, and our results suggest that computational models can be used to predict initiation and development of intraluminal thrombus in MAs as well as provide insights into their role in the pathophysiology of DR.

# 1. Introduction

As of 2019, approximately 463 million adults across the globe were living with diabetes mellitus and this number will rise to 700 million by 2045 [1]. One-third of people with diabetes suffer from diabetic retinopathy (DR), a complication of diabetes that could lead to visual impairment and blindness [2]. Early detection and effective management of DR is essential to protect the sight for diabetic patients [3,4]. Microaneurysms (MAs) are hallmark lesions of DR and their detection in retina is widely used to gauge DR severity and estimate risk of future DR progression. Retinal MAs on fundus photography appear as small, red spots with the size ranging from 10 to 100 µm in diameter [5]. MAs are dilations of capillaries, the smallest vessels in the microvasculature of the retina. Formation of retinal MAs is associated with vessel basement membrane thickening [6], pericyte loss [7] and endothelial cell death [8]. These pathological alterations of the vessel wall can be responsible for the leakage of fluid into the surrounding neural retina layers, precipitating retinal oedema and causing vision loss [9,10]. In addition, rupture of MAs causes leaking of blood into the surrounding tissue, leading to retinal haemorrhage [11]. Since the number of MAs observed on fundus images increases as DR worsens in severity, detection of MAs and evaluation of their leakage status have been used for disease screening and monitoring [12].

Different imaging modalities, such as fundus photography, fluorescein angiography (FA), optical coherence tomography and angiography (OCT and OCTA), as well as adaptive optics scanning laser ophthalmoscopy (AOSLO) have been implemented to detect MAs and examine their leakage status [13]. Nonetheless, each of these modalities has limitations in the imaging of individual MA characteristics. The resolution of fundus photographs is not sufficient to resolve the details of individual MAs, such as their morphologies and perfusion status, i.e. MAs on fundus photographs can be indistinguishable from small intraretinal haemorrhages. FA, an invasive method that uses fluorescent dye to visualize retinal vasculature *in vivo*, is currently the gold standard to evaluate the leakage of MAs. The side effects of the contrast agents, including lightheadedness, nausea and vomiting, as well as rarer instances of allergy and even anaphylaxis, nevertheless limit its application in some DR patients [14,15]. OCT, on the other hand, can capture some, but not all MAs and it also does not address the perfusion status. OCTA can be used to reconstruct three-dimensional images of the perfused retinal vasculature and the associated MAs [16]; however, it is not capable of visualizing vascular leakage and detecting the blood flow when the flow rate is low. Finally, AOSLO can provide ultra-high resolution retinal imaging down to cellular level (approx. 2 µm), one of the highest resolutions of all the available imaging techniques for the human retina. Dubow *et al.* [17] implemented AOSLO to image human MAs *in vivo* and classified the MA's morphologies into six shapes, including focal bulging, saccular, fusiform, mixed saccular/ fusiform, pedunculated and irregular-shaped MAs. Furthermore, AOSLO can dynamically image the blood flow at the capillary level [18].

In order to evaluate the risk of MA leakage without using invasive approaches, previous efforts have been directed toward analysing the sizes and shapes of MAs to explore the association between these characteristics with leakage. A prior clinical study of 173 MAs from 50 eyes showed no correlation between leakage and the lumen diameters of these MAs, suggesting that the size of MAs is not a metric to evaluate the risk of their leakage [19]. A subsequent analytical study demonstrated that the ratio of the size of MA to its feeding vessels can serve as a potential predictor for assessing risk of leakage [20]. Recently, Schreur *et al.* [21] examined 513 MAs from 24 eyes and proposed that the morphology of retinal MAs is a more reliable biomarker than their sizes. Particularly, they found that irregular, fusiform and mixed-shaped MAs are more likely to leak compared to saccular and pedunculated MAs, but the detailed mechanism underlying this association is not fully understood.

Blood clots or thrombi have been detected in various types of aneurysms in the macro-scale, such as cerebral aneurysms [22], abdominal aortic aneurysms [23,24] and dissecting aortic aneurysms [25], where they can either degrade the vessel wall and thus accelerate aneurysm rupture or stabilize the aneurysm by reducing the wall shear stress. However, the role of thrombosis in MAs has not been studied due to the limited resolution ability of standard imaging techniques to determine MA thrombus presence. Ezra *et al.* [20] detected accumulation of von Willebrand factor (VWF) on the dysfunctional vessel wall of MAs with large body-to-neck ratio, which may initiate the adherence of platelets to the vessel wall and thrombus formation. Clinical and histopathological evidence demonstrates that MA turnover, with spontaneous resolution in the retinas of diabetic patients over time, can be associated with thrombosis of the dilated lumen [26]. Dubow *et al.* [17] observed blood cell aggregates in pedunculated and irregular-shaped MAs and proposed that the complex blood flow inside these MAs could increase the risk of endothelial injury and precipitate the formation or enlargement of intraluminal clots. Recent advances in imaging technology through AOSLO enable differentiating the thrombus-filled region

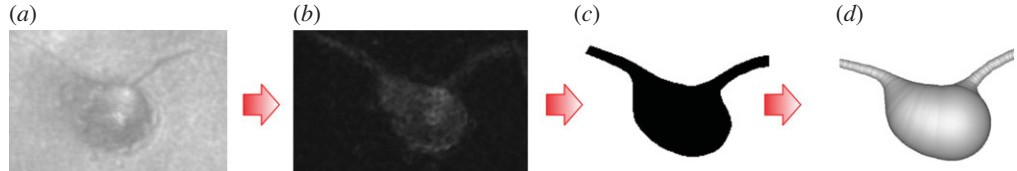

**Figure 1.** Reconstruction of three-dimensional geometry of MAs from AOSLO images. (*a*) The MA is imaged using AOSLO multiply scattered light (MSL) imaging modality. (*b*) The perfusion map of the corresponding MA is generated by analysing a sequence of MSL images using pixel-by-pixel standard deviation method. The bright region in the map highlights the region with flowing blood. (*c*) A binary mask of the MA and its feeding and draining vessels are created manually based on (*a*) and (*b*) using ImageJ [29]. (*d*) Three-dimensional models of the MA are generated under the assumption that the MA geometries are rotationally symmetric.

from the perfused region in MAs. Bernabeu *et al.* [27] reported that on AOSLO imaging thrombi are more frequently observed in saccular-shaped MAs compared to fusiform-shaped MAs.

Although retinal imaging now enables us to reconstruct the geometries of individual MAs and to perform computational fluid dynamics (CFD) studies to quantify the haemodynamic metrics, such as shear rate and wall shear stress [27], no computational studies have been performed to predict the initiation and formation of a thrombus in MAs. The ability to predict the course of thrombus development in MAs and thereby potentially better understand the overall risk of DR worsening might help ophthalmologists to monitor the progression of DR and guide appropriate timing for clinical intervention. In this study, we employ a particle-continuum model based on a force coupling method (FCM) to quantitatively predict the presence of a thrombus and its extent in MAs. This model implements a two-way coupling between the platelets (treated as semi-rigid spherical particles) and the background continuum blood flow, thereby allowing us to simulate the platelet flowing and aggregating in MAs with various morphologies under different haemodynamic conditions. The three-dimensional MA geometries used in the simulations are reconstructed based on retinal images obtained using the AOSLO imaging technique. The rest of the article is organized as follows. In the Model and methods section, we introduce the method we used to reconstruct the three-dimensional MA geometries based on the AOSLO retinal images. We also introduce the FCM method we employed to simulate the platelet aggregation in the MAs. In the Results section, we first investigate the effect of the MA shape on the intraluminal thrombus formation. Then, we vary the blood flow rates in MAs and examine how haemodynamics affects the growth of thrombi in MAs. In the Discussion section, we make comparisons between the model predictions and clinical observations, and explore the clinical implications of the simulation results.

# 2. Model and methods

## 2.1. Clinical image processing and microaneurysm geometry reconstruction

Four MAs from the eyes of adult study participants with diabetes (type 1 or 2) underwent AOSLO imaging. All MAs were located within approximately 20° from the foveal centre. The AOSLO system used has been previously described in detail by Lu *et al.* [28]. This system uses confocal and multiply scattered light (MSL) imaging modes, and achieves a field size of approximately $1.75° \times 1.75°$ with lateral resolution of approximately 2.5 μm on the retina. Moreover, 75-frame videos of each MA were aligned and averaged (Matlab, The MathWorks, Inc., Natick, MA, USA). The magnification factor on AOSLO images was determined by eye axial length measurement or derived from the spherical equivalent of the eye. For this exploratory study, we included MAs with high-quality AOSLO images where the two-dimensional MSL images and corresponding perfusion maps provided sufficient details to identify the full extent of MAs' bodies and their parenting vessels' boundaries. The body and feeding/draining vessels of these MAs were manually segmented based on MSL images and their corresponding perfusion maps by using ImageJ [29]. As shown in figure 1, the MA outline is adjusted based on both AOSLO MSL (figure 1*a*) and the corresponding 'perfusion map' (figure 1*b*), which is obtained by calculating the standard deviation of a sequence of AOSLO images. The four reconstructed MA geometries are illustrated in figure 2*a*–*d*. The segmented MA (figure 1*c*) or the mask, is used to generate the three-dimensional model for simulation (figure 1*d*) under the assumption that the MA geometries are rotationally symmetric. More details of this method can be found in [27].

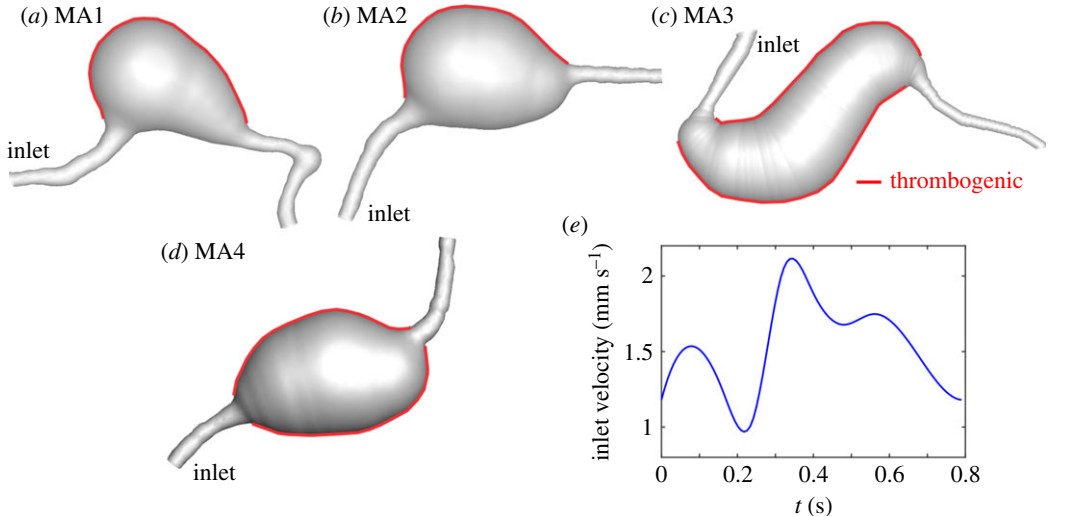

**Figure 2.** Four MA geometries are selected to perform the simulations, including (*a,b*) saccular and (*c,d*) fusiform shapes. The inner surface of the MAs is assumed to be thrombogenic (highlighted by red colour). (*e*) A pulsatile flow profile with a mean velocity of $U_0 = 1.6$ mm s$^{-1}$, following the measurements in [18], is imposed at the inlet and a zero pressure condition is prescribed at the outlet.

## 2.2. Platelet aggregation model

Numerous computational studies were conducted to simulate platelet transport, activation and aggregation under both physiological and pathological conditions [24,30–41], but few of them have focused on thrombus formation in MAs. In this work, we simulate platelets aggregation on thrombogenic surfaces inside MAs using FCM [42] integrated with the spectral element method (SEM) [43]. SEM is employed to solve the flow field on a fixed Eulerian grid, whereas FCM is used to describe the motion of platelets and their (bi-directional) interactions with blood flow. This methodology has been successfully used in modelling platelet aggregation in venules [44], stenotic channels [45] and intramural dissections [25]. The Carreau–Yasuda rheology model parametrized for human blood is implemented to capture the shear-thinning behaviour [46]. The velocity profile at the inlet was assumed to be parabolic for a given centreline peak velocity. Since the specific blood flow velocity for each MA was not measured, we adopted the measurement of blood flow velocities in parafoveal capillaries by de Castro *et al.* [18], which has a mean capillary velocity of $U_0 = 1.6$ mm s$^{-1}$ (figure 2*e*).

In our simulations, platelets with a radius $r_p = 1.5$ μm exhibit four different biostates, namely *passive*, *triggered*, *activated* or *adhered*. In passive or triggered states, platelets are non-adhesive. Once platelets in the passive states interact with activated platelets or the thrombogenic area, they switch to a triggered state. Triggered platelets become activated after a delay time $\tau = 0.1$–$0.2$ s [44]. The activation delay time is selected based on the *in vivo* study performed by Begent & Born [47], who obtained quantitative data on thrombus growth rates for a range of blood flow rates. Their study implied that there is a delay between each platelet's close encounter with a thrombus and its development of ability to adhere to that thrombus. This activation delay time of the platelets was estimated to be 0.1–0.2 s [44]. The activated platelets are adhesive and they can adhere to other activated platelets or thrombogenic wall. The activated platelet particles are considered to be 'adhered' to the clot when their moving distance within one cardiac cycle is less than 1/10 of their radius.

Since the detailed mechanism of thrombosis in MAs is not well understood, we employed a phenomenological adhesion model where the interactions between the platelets as well as between the platelets and the vessel wall are described by a Morse potential [45]. The interaction force is shear-rate dependent and the force values were validated against the data from four independent experimental studies, including two *in vivo* [25,47] and two *in vitro* [48,49] experiments, which measured platelet aggregation at different shear rates. Detailed information of the FCM model can be found in the electronic supplementary material, and our prior work in [25,45]. To better illustrate the thrombi and their structures in MAs, we convert the particle-based simulation results to a continuum representation of the adhered platelets by computing the volume fraction of the thrombus $\phi$, evaluated by $\phi(\mathbf{x}, t) = \sum_{n=1}^{N} \mathcal{V}_p{}^n \Delta(\mathbf{x} - \mathbf{Y}^n(t))$, where $\mathcal{V}_p{}^n$ is the volume of the platelets $n$, $\mathbf{x}$ is the position of the background Eulerian grid, $\mathbf{Y}^n$ is the position of the platelet $n$, and $t$ is simulation time. The volume

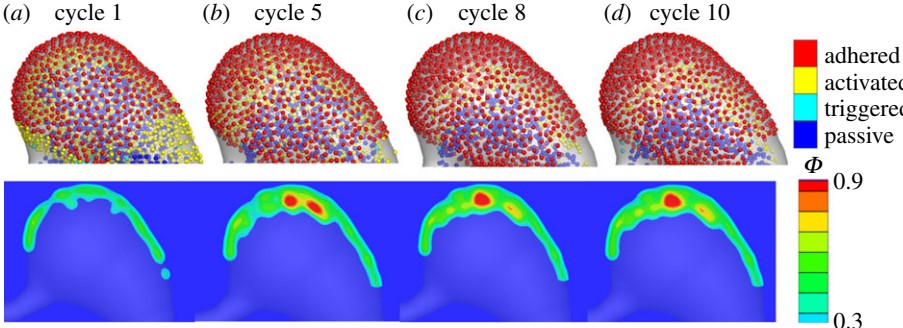

**Figure 3.** Simulation results of platelet aggregation in MA1. (*a–d*) Four snapshots of platelet aggregation inside the dilated lumen of MA1 at cycle 1, 5, 8 and 10, respectively. Top: FCM simulation results. Bottom: conversion of FCM simulation results to volume fraction $\phi$.

fraction is smoothed by a Gaussian distribution kernel $\Delta(\mathbf{x} - \mathbf{Y^n})$ with the standard deviation of the kernel, $\sigma$, related to particle radius through $\sigma = r_p / \sqrt{\pi}$.

As the development of thrombi in retinal MAs can be a week-long or even month-long process, we simulate a final-value problem in lieu of the original initial-value problem by initially placing passive platelet particles uniformly inside the dilated lumen of MAs such that once the simulation starts, there are sufficient platelet particles to form aggregates. Meanwhile, new platelet particles are inserted from the inlet throughout the cardiac cycle. In the simulation, the platelet particles that do not adhere to existing aggregates or the thrombogenic wall will flow out of the MAs. This assumption is physiologically relevant as both the pre-aneurysmal lesions and the enlarging aneurysms always contain blood, and thus are filled with large number of platelets (as well as erythrocytes and leukocytes). This method can largely accelerate the simulation of thrombus formation and it has been successfully implemented in our previous work to predict the development of thrombi in aortic dissecting aneurysms in a mouse model [25].

# 3. Results

## 3.1. Platelet aggregation in microaneurysms

In this section, we simulate the platelet aggregation in the dilated lumen of MAs. First, we model platelet aggregation in a saccular-shaped MA as illustrated in figure 2*a*. Our FCM simulation results in figure 3*a* (top) show that after one cardiac cycle, the platelets close to the aneurysm wall become activated and adhered to the wall where the wall shear stress is low (see plots of wall shear stress in electronic supplementary material, figure S1). As the simulation continues, an increased number of activated platelets adhere to the aggregates (figure 3*b–d*(top)). We run the simulation until the platelet aggregates become stable. Next, we convert the FCM simulation results to the volume fractions of platelet aggregates. Figure 3*a–d*(bottom) shows that a thin layer of thrombus is formed close to the wall at the beginning of the simulation and then the thrombus gradually develops towards the blood flow. After simulating for eight cardiac cycles, the thrombus is stabilized and mainly located around one side of the aneurysm. Quantitative data for the growth of the thrombus with respect to the cardiac cycles can be found in electronic supplementary material, figure S2. Inferred from the velocity field in the dilated lumen of MA1 (figure 4*a*), the growth of the thrombus is eventually inhibited by the blood flow when the flow velocity is larger than approximately 0.032 mm s$^{-1}$. These simulation results suggest that the local haemodynamics in the dilated lumen of the MA is essential to the development of the thrombus. It is noted that our model prediction of size and shape of the thrombus inside the dilated lumen of MA1 (figure 4*b*) is consistent with observations from AOSLO images and the perfusion map (figure 4*c–d* and table 1), where the non-perfused area in the dilated lumen of the MA is assumed to be thrombosed.

## 3.2. Effect of microaneurysm's shape on intraluminal thrombus formation

In this section, we perform FCM simulations for three additional MAs with varying morphologies (figure 2*b–d*) to investigate the effect of the shape of MAs on the thrombus formation. The mean inlet velocity $U_0$

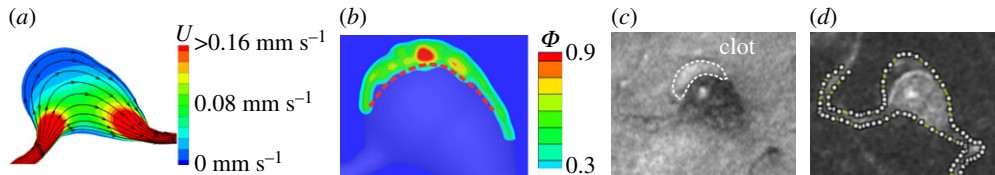

**Figure 4.** Comparison of the simulation results to the experimental observations. (*a*) Velocity field and streamlines in the cross-section of MA1 computed at an inlet velocity of $U_0 = 1.6$ mm s$^{-1}$. (*b*) The platelet aggregation inside the dilated lumen of MA1 at cycle 10 expressed in terms of the volume fraction $\phi$. (*c*) AOSLO image of MA1, and (*d*) the corresponding perfusion map. The red dotted line in (*b*) signifies the boundary between the flowing blood and thrombus observed in (*c*). The white dotted line in (*c*) highlights the boundary of the thrombus. The white dotted line in (*d*) highlights the boundary of the MA1. (*c,d*) are adapted from [27].

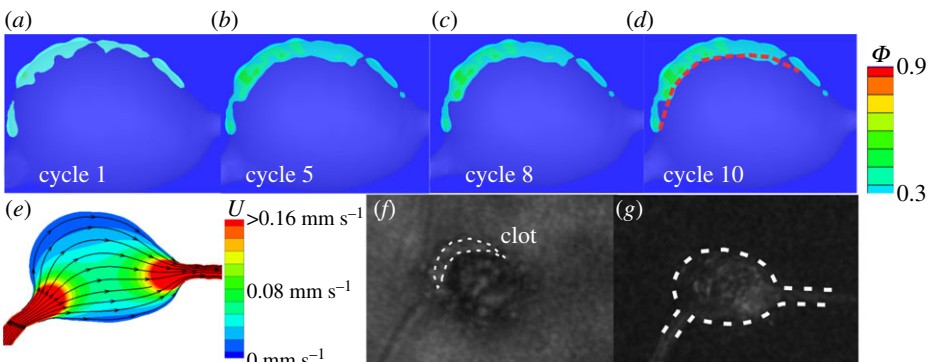

**Figure 5.** Simulation results of platelet aggregation in MA2 and comparison to the experimental observations. (*a–d*) Four snapshots of platelet aggregation inside the dilated lumen of MA2 at cycle 1, 5, 8 and 10, respectively. (*e*) Velocity field and streamlines in the cross-section of MA2 computed at an inlet velocity of $U_0 = 1.6$ mm s$^{-1}$. (*f*) AOSLO image of MA2 and (*g*) the corresponding perfusion map. The red dotted line in (*d*) signifies the boundary between the flowing blood and thrombus observed in (*f*). The white dotted line in (*f*) highlights the boundary of the thrombus whereas the white dotted line in (*g*) highlights the boundary of the MA2.

**Table 1.** Geometric parameters of the examined MAs measured from AOSLO images.

|  | inlet diameter | projected area | volume | asymmetric ratio | clot presence |
|---|---|---|---|---|---|
| MA1 | 5.8 μm | 2734.4 μm$^2$ | 62772.2 μm$^3$ | 2.17 | yes |
| MA2 | 7.0 μm | 1849.4 μm$^2$ | 66990.1 μm$^3$ | 1.47 | yes |
| MA3 | 10.2 μm | 5454.9 μm$^2$ | 155713.1 μm$^3$ | 1.09 | no |
| MA4 | 6.2 μm | 2658.3 μm$^2$ | 74927.8 μm$^3$ | 1.09 | no |

for the simulations of these three new geometries is maintained at 1.6 mm s$^{-1}$. First, we simulate the evolution of a thrombus in another saccular-shaped MA (MA2), as shown in figure 2*b*. Our simulation results in figure 5*a–d* illustrate a process of thrombus development similar to that of MA1 where the platelet aggregate is initiated from one side of the aneurysm and grows towards the blood flow until the blood flow inhibits its expansion. The growth of the thrombus with respect to the cardio cycles can be found in electronic supplementary material, figure S2. However, the thrombus formed in MA2 is thinner and more porous compared to that in MA1 due to the altered shapes of MAs. Figure 5*d* shows that the size and shape of the thrombus predicted by the model is also in agreement with observations from the AOSLO images and the perfusion map (figure 5*f–g* and table 1).

Second, we simulate the thrombus formation in a fusiform-shaped MA as illustrated in figure 2*c*. Figure 6*e* shows that the flow velocity in the dilated lumen of MA3 is much higher compared to those of two saccular-shaped MAs in figure 4*a* and 5*e*. As a result, our model predicts that only a very thin layer of thrombus is formed close to the vessel wall of MA3 (see figure 6*a–d* and table 1), which agrees with the fact that no clots could be observed from the AOSLO images (figure 6*f*). The perfusion map for MA3 in figure 6*g* shows that the dilated lumen of MA3 is almost fully perfused, implying that no or minimal thrombus is formed in MA3.

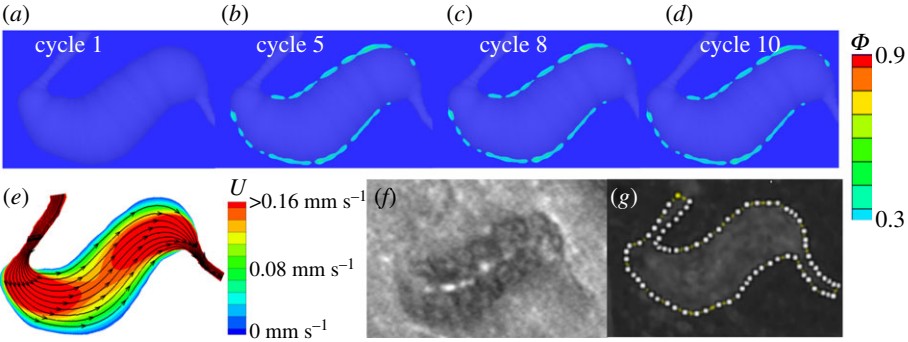

**Figure 6.** Simulation results of platelet aggregation in MA3 and comparison to the experimental observations. (*a–d*) Four snapshots of platelet aggregation inside the dilated lumen of MA3 at cycle 1, 5, 8 and 10, respectively. (*e*) Velocity field and streamlines in the cross-section of MA3 computed at inlet velocity $U_0 = 1.6$ mm s$^{-1}$. (*f*) AOSLO image of MA3 and (*g*) the corresponding perfusion map. The white dotted line in (*g*) highlights the boundary of the MA3. (*f–g*) are adopted from [27].

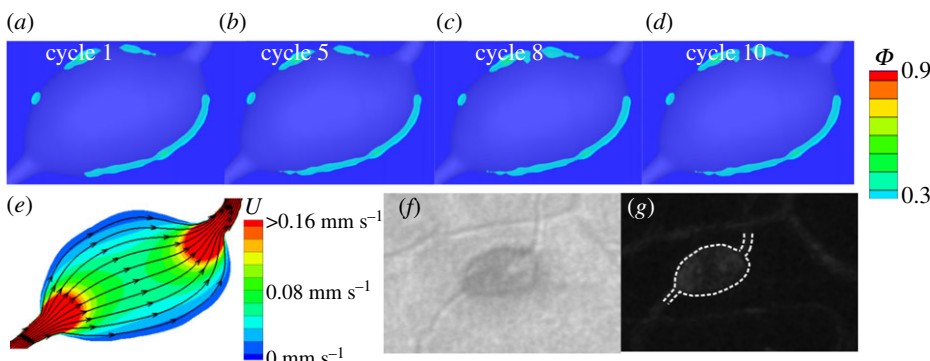

**Figure 7.** Simulation results of platelet aggregation in MA4 and comparison to the experimental observations. (*a–d*) Four snapshots of platelet aggregation inside the dilated lumen of MA4 at cycle 1, 5, 8 and 10, respectively. (*e*) Velocity field and streamlines in the cross-section of MA4 computed at inlet velocity $U_0 = 1.6$ mm s$^{-1}$. (*f*) AOSLO image of MA4 and (*g*) the corresponding perfusion map. The white dotted line in (*g*) highlights the boundary of the MA4.

Next, we examine another fusiform-shaped MA4 illustrated in figure 2*d*. Our simulation results in figure 7*a–d* show that at the beginning of the simulation, two thin layers of thrombus with low volume fraction are formed in regions adjacent to the vessel wall where the blood velocity is low and they do not grow toward the blood flow as the simulation continues. This result is also comparable with the observations from the corresponding AOSLO images and perfusion maps (figure 7*f–g* and table 1). Our simulation results above demonstrate that our model is capable of predicting the possible formation of thrombus for newly detected MAs from medical images. Moreover, we find that thrombosis is more likely to occur in saccular-shaped MAs than fusiform-shaped MAs, consistent with the observations reported in [27].

Next, we quantify the geometric parameters for the four examined MAs, namely the projected area of the dilated lumen of MAs (the area of the dilated lumen in the mask), the volume of dilated lumen of MAs and dilated lumen asymmetric ratio (AR), which are listed in table 1. AR is computed as the ratio of two areas separated by the centreline on mask of the MAs (AR≥1), following the definition in [27]. Based on these parameters and our quantified simulation results in figure 9, we find that the extent of thrombus formation in MAs is not correlated with the projected area and volume of the dilated lumen, suggesting that the size of the dilated lumen is not a metric to predict the intraluminal thrombus development. However, we note that the thrombus formed in MA1 (AR = 2.18) is more prominent than that in MA2 (AR = 1.47). The extent of thrombi becomes less prominent as the AR decreases to AR = 1.09 for MA3 and MA4. These findings confirm previous clinical observations that thrombi are more likely to be present in MAs with high AR [27]. Our results also could suggest a potential role for intraluminal thrombosis in preventing MA leakage if the previous findings that saccular-shaped MAs pose a lower risk of leakage than the fusiform-shaped MAs [21] are validated.

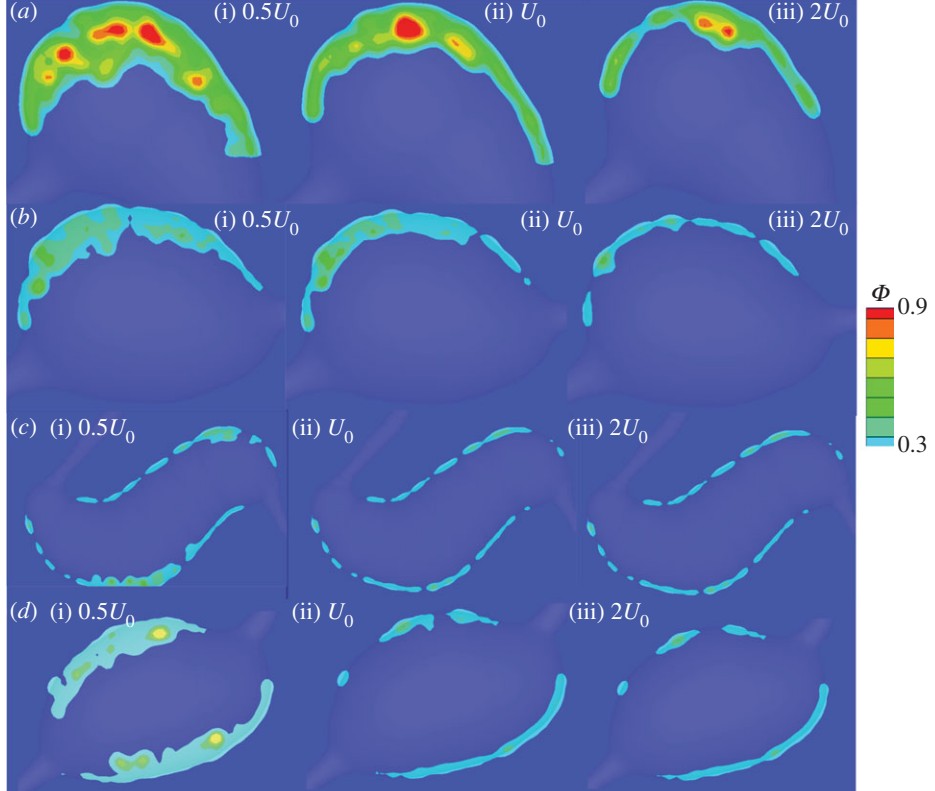

**Figure 8.** Comparison of the platelet aggregation at three different inlet velocities, namely half of the reference velocity (left), the reference velocity (middle), and twice of the reference velocity (right), for the four selected MA geometries (a–d). The snapshots of platelet aggregation inside the dilated lumen of MAs are captured at the 10th cardiac cycle.

## 3.3. Dependence of growth of thrombus on haemodynamic conditions

The inlet blood flow velocity employed in our simulations was adopted from a separate measurement of parafoveal capillaries [18] and it was not directly measured for MAs studied in the current work. This velocity can be varied in individual patients with diabetes. In addition, the retinal haemodynamics progressively change due to capillary dropout [50,51], vasodilation [52,53] and the concomitant altered retinal perfusion and autoregulation in DR. Although the existing literature has reported both increased and decreased retinal blood flows in eyes with DR in multiple studies using different measurement techniques, see a review [54], the current consensus is that total retinal blood flow decreases initially in early DR and then increases due to vascular shunting in advanced DR. Vascular density decreases with increased DR severity [55]. To account for a range of blood flow effects on thrombus development in MAs, we vary the inlet blood velocity by changing the mean inlet velocity of $U_0 = 1.6$ mm s$^{-1}$ to $0.5U_0$ and $2U_0$, respectively, for the four MAs and investigate how the variations in the inlet blood velocity affect the growth of intraluminal thrombus.

Our simulation results for the two saccular-shaped MAs (figure 8a,b) show that thrombi formed in MA1 and MA2 develop to greater extents in the dilated lumen when the inlet velocity is reduced by half and become depleted when the inlet velocity is doubled. Figure 9a shows that the percentages of thrombus areas in the projected areas of dilated lumen undergo drastic drops for MA1 and MA2. The corresponding percentages of volume occupied by the thrombi in the dilated lumen of MAs (figure 9b) also decrease with increased inlet velocities. These results suggest that blood flow rates in the feeding vessels of saccular-shaped MAs could play an important role in determining the shape and size of the intraluminal thrombus. On the other hand, for the fusiform-shaped MA3, as shown in figure 8c, the extent of thrombus is not substantially different at these three inlet blood velocities. The area and volume percentages of the thrombus experience a slower drop compared to the two saccular-shaped MAs (figure 9a,b). The simulation results for MA4 in figure 8d and figure 9 show that while an increase of the inlet velocity from $U_0$ to $2U_0$ does not change the extent of thrombi, a decrease in inlet velocity from $U_0$ to $0.5U_0$ substantially boosts the growth of the thrombus. These

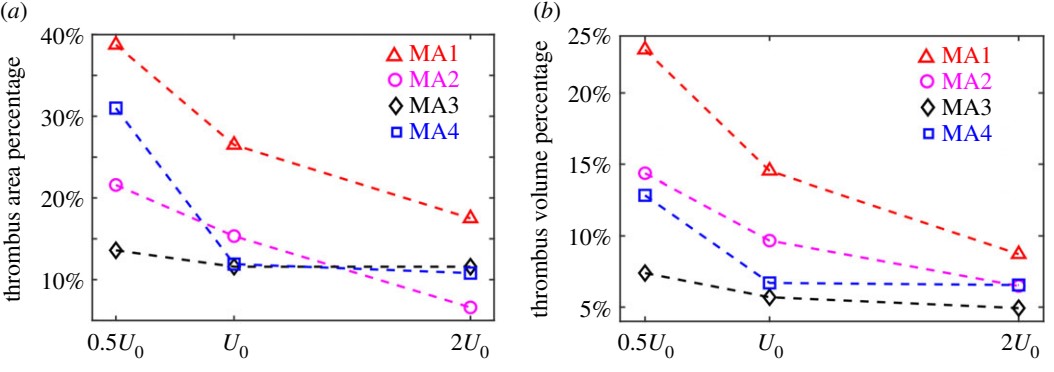

**Figure 9.** (*a*) Percentage of area occupied by the adhered platelets in the projected area of dilated lumens of MAs and (*b*) percentage of volume occupied by the adhered platelets in the volume of the dilated lumens of MAs at three different inlet velocities for the four selected MA geometries.

results suggest that the changes in blood flow in the retinal microvasculature could contribute to the thrombus formation in at least some fusiform-shaped MAs.

## 4. Discussion

MAs are one of the earliest clinical signs detected by the routine fundus examination for DR [5]. Leakage or rupture of MAs result in retinal oedema and haemorrhage, respectively, which may, in turn, lead to functional visual impairment. Therefore, accurate evaluation of the stability of MAs could contribute to improved management of DR and better determination of the best timing for treatment and intervention. Three-dimensional luminal surfaces of patient-specific MAs have been reconstructed based on the retinal images in order to quantify the haemodynamic metrics inside MAs, such as shear rate and wall shear stress. Implementation of AOSLO techniques allows visualization of thrombi in MAs, but how the intraluminal thrombus affects the stability of its residing MA is not clear. Data reported from recent clinical studies [21,27] implies that thrombosis in MAs could be associated with reductions in leakage. In this study, we simulate the platelet aggregation in the dilated lumen of MAs by coupling SEM with FCM. The geometries of patient-specific MAs are reconstructed based on the AOSLO images. Our simulation results demonstrate the strong effects of local haemodynamics in the dilated lumen of MAs on the final size and shape of intraluminal thrombus. The initiation of thrombus starts from the regions where the blood flow is stagnant and the wall shear stress is low (see electronic supplementary material, figure S1 for wall shear stress in the four examined MAs) whereas the growth of the thrombus is limited eventually in local areas of faster blood flow. These findings for micro-scale aneurysms are consistent with experimental and computational studies for large-scale aneurysms, such as aortic dissecting aneurysm [25,56]. Our simulation results also show that our model predictions of the size and shape of the intraluminal thrombus are consistent with experimental observations, demonstrating the capability of our model to predict the presence of thrombus for newly detected MAs.

Advances in the retinal imaging techniques in the last decade enable the classification of the morphologies of individual MAs [17], which may be used as a metric to evaluate the risk of MA leakage. A recent clinical study [21] reported that irregular, fusiform and mixed-shaped MAs tended to leak more frequently than saccular-shaped MAs, but the underlying mechanism is not well understood. Our simulation results demonstrate that under the same physiologically relevant inlet flow boundary condition, thrombosis is more prominent in the two examined saccular-shaped MAs than the two examined fusiform-shaped MAs, consistent with our previous observational study [27]. These findings support the hypothesis that intraluminal thrombus could be associated with decreased tendency for MAs to leak, or possibly play a role in preventing the MAs from leaking. In addition, our simulation results show that the size of a MA does not correlate with thrombus formation. Thrombus formation is more pronounced in MAs with increased AR, which is consistent with experimental observations reported in [27]. It is noted that the above findings were obtained under the assumption that the same physiologically relevant inlet blood velocity is imposed at the parenting vessels of the examined MAs. We also examine how the changes of retinal perfusion affect the

development of thrombus in MAs. As shown in figures 8 and 9, the variations in the inlet blood flow velocity could substantially change the development of the thrombi in MAs. In particular, a decrease in blood flow velocity could increase the growth of thrombus in the fusiform-shaped MAs, which did not appear to have thrombus either clinically or upon modelling at the physiologically relevant inlet blood velocity. This finding suggests that patient-specific blood flow rate for individual MAs may be necessary to enable more precise estimation of the intraluminal thrombus formation.

The motivation of this study is to introduce a computational model that can quantitatively predict thrombus formation and its extent in MAs with varying shapes under different haemodynamic conditions. It is noted that we did not include other blood cells such as red blood cells in our simulations and thus we cannot consider their interaction with platelets, which could affect platelet transport, particularly in the lumen of the MAs. A more detailed model with explicit representations of the red blood cells and platelets, such as [57–62], could be used to explore this effect. See our ongoing work to address this problem in electronic supplementary material, figure S3. Another potential extension of this current model is to introduce the flow–structure interaction to describe the interaction between flowing blood and vessel wall and evaluate the deformation of the wall and its propensity to rupture. However, it requires the knowledge of the mechanical properties of the damaged vessel wall for model calibration, which is still not well studied. In addition, we assume based on previous vascular modelling work [25] that the inner surface of MAs is thrombogenic but the underlying mechanism for platelet aggregation inside MAs is not fully clear yet. This assumption could cause overestimated formation of thrombus inside MAs.

The good agreement between clinically apparent thrombus and our computational modelling results for diabetic MAs suggests that this computational approach may prove useful to understand thrombus formation in microvascular pathology. Accurate prediction of thrombus initiation and progression in MAs using computational simulation could facilitate the determination of the best timing for therapeutic intervention. Currently, there are no clinical data to directly associate the intraluminal MA thrombus with either positive or negative clinical outcomes. The clinical disappearance of some MAs could result from the full thrombosis of their dilated lumens [26]. In addition, fully thrombosed MAs with no perfusion may be less likely to leak fluid into the neural retina. However, we cannot rule out the possibility that intraluminal thrombus may accelerate the leakage or rupture of some MAs and it is also possible that MAs develop thrombus due to endothelial cell injury that is likely to cause either past or current leakage. The computational modelling methods presented here could be used in a future longitudinal study with follow-up of MAs from a large cohort of patients to dramatically improve our understanding of the role of thrombus in the pathophysiology of MAs in the diabetic eyes.

Data accessibility. All data and codes used in this manuscript are publicly available on GitHub [63].

Authors' contributions. All authors contributed to the writing and revision of the manuscript.

Competing interests. We declare we have no competing interest.

Funding. H.L., X.Z., A.Y. and G.E.K. acknowledge the support from NIH grant nos. U01 HL1163232 and U01 HL142518. K.S. and J.K.S. acknowledge the support by NEI 25R01EY024702-04 as well as grants from Research to Prevent Blindness, JDRF 3-SRA-2014-264-M-R, and the Massachusetts Lions Eye Research Fund. M.O.B. is supported by grants from EPSRC (EP/R029598/1, EP/R021600/1), Fondation Leducq (17 CVD 03), and the European Union's Horizon 2020 research and innovation programme under grant agreement no. 801423.

Acknowledgements. H.L. and X.Z. thank Dr Zhicheng Wang and Dr Shenzen Cai for helpful discussion. High performance computing resources were provided by the Center for Computation and Visualization at Brown University and the Extreme Science and Engineering Discovery Environment (XSEDE), which is supported by National Science Foundation grant nos. ACI-1053575, TG-DMS140007 and TG-MCB190045.

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
