## [Reviewer comments · Royal Society Open Science]

Review History

RSOS-201102.R0 (Original submission)

Review form: Reviewer 1 (Xiao Liu)

Is the manuscript scientifically sound in its present form?

Yes

Are the interpretations and conclusions justified by the results?

Yes

Is the language acceptable?

Yes

Do you have any ethical concerns with this paper?

Yes

Have you any concerns about statistical analyses in this paper?

No

Recommendation?

Accept as is

Comments to the Author(s)

The author made a series of amendments to my questions, which sounds a lot more reasonable, and has a certain basis. Although I still think that red blood cells and arterial deformation are important factors to consider, this article makes sense as a preliminary study of microarteries.

Review form: Reviewer 2 (Gabor Zavodszky)**Is the manuscript scientifically sound in its present form?**

Yes

Are the interpretations and conclusions justified by the results?

Yes

Is the language acceptable?

Yes

Do you have any ethical concerns with this paper?

No

Have you any concerns about statistical analyses in this paper?

No

Recommendation?

Accept as is

Comments to the Author(s)

The authors have addressed my comments in a satisfactory way. I suggest to accept the manuscript for publication.

Review form: Reviewer 3**Is the manuscript scientifically sound in its present form?**

Yes

Are the interpretations and conclusions justified by the results?

Yes

Is the language acceptable?

Yes

Do you have any ethical concerns with this paper?

No

Have you any concerns about statistical analyses in this paper?

No

Recommendation?

Accept as is

Comments to the Author(s)

The Authors satisfactorily addressed all the comments.

Decision letter (RSOS-201102.R0)

Dear Dr Li:

It is a pleasure to accept your manuscript entitled "Predictive modeling of thrombus formation in diabetic retinal microaneurysms" in its current form for publication in Royal Society Open Science. The comments of the reviewer(s) who reviewed your manuscript are included at the foot of this letter.

on behalf of Dr Kenta Ishimoto (Associate Editor) and Professor R. Kerry Rowe (Subject Editor).

Reviewer(s)' Comments to Author:

Reviewer: 1

Comments to the Author(s)

The author made a series of amendments to my questions, which sounds a lot more reasonable, and has a certain basis. Although I still think that red blood cells and arterial deformation are important factors to consider, this article makes sense as a preliminary study of microarteries.

Reviewer: 2

Comments to the Author(s)

The authors have addressed my comments in a satisfactory way. I suggest to accept the manuscript for publication.

Reviewer: 3

Comments to the Author(s)

The Authors satisfactorily addressed all the comments.
